# Evolution of Public Health Prevention of Leptospirosis in a One Health Perspective: The Example of Mahasarakham Province (Thailand)

**DOI:** 10.3390/tropicalmed6030168

**Published:** 2021-09-17

**Authors:** Jaruwan Viroj, Claire Lajaunie, Serge Morand

**Affiliations:** 1Faculty of Public Health, Mahasarakham University, Maha Sarakham 44150, Thailand; 2Inserm, UMR LPED (IRD, Aix-Marseille Université), F-13005 Marseille, France; claire.lajaunie@inserm.fr; 3Strathclyde Centre for Environmental Law and Governance (SCELG), Law School, Strathclyde University, Glasgow G4 0RE, UK; 4CIRAD, UMR ASTRE (Animals, Health, Territories, Risks and Ecosystems), F-34398 Montpellier, France; serge.morand@umontpellier.fr; 5Faculty of Veterinary Technology, Kasetsart University, Bangkok 10900, Thailand

**Keywords:** leptospirosis, One Health, prevention and control, Thailand, public policy implementation

## Abstract

Leptospirosis is an endemic disease with moderate to high incidence in Mahasarakham province, Thailand. The present study was designed to assess the policy implementation mission regarding leptospirosis prevention and control from the national level to the local administrative levels, through a One Health perspective. A qualitative study was conducted, using documentation review, individual in-depth interviews with public health officers, local government officers, livestock officers who developed policy implementation tools or have responsibilities in leptospirosis prevention and control. The results show that Thailand has progressively developed a leptospirosis prevention and control policy framework at the national level, transferring the responsibility of its implementation to the local level. The province of Mahasarakham has decided to foster cooperation in leptospirosis prevention and control at the local level. However, there are insufficient linkages between provincial, district and sub-district departments to ensure comprehensive disease prevention activities at the local level concerning leptospirosis patients and the whole population.

## 1. Introduction

Leptospirosis is an important zoonotic disease and is considered as one of the most widespread zoonoses in the world [1,2,3,4].The disease is caused by several spirochete species of the genus *Leptospira* [5]. Infected patients can develop severe symptoms characterized by hepatic, renal, or hemorrhagic manifestations which can lead to death [6,7].

Thailand is located in the tropical zone with important soil moisture and high rainfalls during the wet season, which favors leptospire transmission. The first isolation of leptospires from human patients in Thailand was reported in 1942 [8]. More than 70,000 cases have been recorded since the emergence of this disease in Thailand [4]. Leptospirosis incidence shows a strong seasonality with a high incidence during the wet season [9].

The province of Mahasarakham is situated in the Isan part (North East) of Thailand characterized by a moderate to high leptospirosis incidence. In Mahasarakham, the first leptospirosis case in a human patient was reported in 1996. The province, as most other rural areas of Thailand, is characterized by its livestock management. Cattle and buffaloes graze in harvested rice fields, which may favor the spread of *Leptospira* in the environment, while pigs are confined in pigsties of varying sizes, which should result in lower leptospire transmission. Accordingly, the dominant *Leptospira* serovar Shermani found in Thailand showed a high variability in prevalence across humans (with a prevalence of 23.7%), buffaloes and cattle (28.1% and 24.8%, respectively), and in pigs (11.3%) suggesting different transmission pathways between humans and livestock [10]. Sharing of habitat and water sources between humans and animals has been identified as a risk factor for disease exposure [11].

During the period 2004 to 2014, leptospirosis cases were reported with an average annual incidence rate of 7.97 cases per 100,000 persons in Mahasarakham Province. For more than two decades, the Ministry of Public Health and the Mahasarakham Provincial Public Health Office have established and developed policy implementation of leptospirosis prevention and control. However, despite some success, leptospirosis is still an endemic disease in the province.

At the local level, the public health department and livestock department are responsible for prevention and control of leptospirosis. The World Health Organization (WHO) has highlighted the importance of the One Health approach to design and implement programs involving the collaboration of multiple sectors communicating and working together [12] and connecting health threats at the human–animal-ecosystems interfaces to achieve better public health outcomes [13]. Collaboration between veterinarians dealing with livestock and wild animal populations, ecologists and public health experts enhances prevention and control [14].

Thailand has integrated the One Health approach into the Thailand National Strategic Plan for Emerging Infectious Disease Preparedness, Prevention and Response [15]. Thailand has also played an important role in promoting and encouraging the use of the One Health approach within the Association of South-East Nations (ASEAN) notably under the South East Asia One Health University Network (SEAOHUN) [16] and in driving the development of the One Health approach at the regional level.

Leptospirosis is still considered a health issue in the province of Mahasarakham and was chosen as a pilot province by the Ministry of Public Health of Thailand for leptospirosis surveillance and control. The aim of this qualitative study is to analyze the public health strategies adopted over time by the province to implement and complete national policies and strategies for leptospirosis prevention and control over time. The objective of this analysis is to determine the necessary elements of policy, particularly those supposed to follow the One Health approach, which will help to enhance the efficiency of the prevention, detection and response to leptospirosis outbreaks.

## 2. Materials and Methods

### 2.1. Study Design

This qualitative longitudinal study was conducted to describe the different steps of the implementation of policies for leptospirosis prevention and control, since the identification of the first case of the disease in the province in 1996. It associates the analysis of primary documents provided by the Public health Department Office and in-depth interviews of Public health officers. This study pays a specific attention to the introduction of the One Health aspect in the measures adopted at the provincial level.

### 2.2. Study Population

The study population included the head of the Department of Disease Control, the officer of Coordination Unit for One Health, public health officers, livestock officer and local government officers of Mahasarakham, to describe the factors that have enabled or constrained the planning and implementation of policies regarding leptospirosis prevention and control. Initial findings were integrated back into the investigation and were used to guide further exploration and analysis of key issues as they emerged. The study population is given in Table 1.

## 3. Data Collection

### 3.1. Data Collection

Data collection consisted of two steps: (i) review primary documents; (ii) data collection among officers from different administrative departments.

#### 3.1.1. Steps of Data Collection

First, we reviewed primary documents obtained from the Public Health Department (provincial reports, meeting reports, command documents, handbooks). Access to those non-publicly available documents was granted by the Deputy Head of the Department of Disease Control of Mahasarakham Provincial Public Health Office. We also used 18 secondary documents: journal articles, presentation handouts and internet information published from 1990 to 30 July 2017. All the documents were selected following two criteria:

(i) documents relating to leptospirosis prevention and control in Mahasarakham, Thailand and ASEAN;

(ii) documents relating to One Health implementation in Mahasarakham, in Thailand (national level) and ASEAN (regional level).

Secondly, researchers collected data among officers from different administrative departments at various decision-making levels using individual in-depth interviews, and participant observations (during meetings and disease investigations).

#### 3.1.2. In-Depth and Structured Interviews

In the thirteen districts of Mahasarakham Province, 13 Public health officers within the subdistricts characterized by high leptospirosis incidence were selected by purposive sampling.

The selected interviewees were the head of the Department of Disease Control, the Officer of the Coordination Unit for One Health, three Provincial Public Health officers, three District Public Health officers, three local government officers, two District Livestock officers.

Each of the interviewees were asked on their position and workplace and the interviews were all dated and indicated where they took place. The in-depth interviews were conducted using flexible topic guides with open-ended questions that allowed participants to express their views on the implementation of leptospirosis prevention and control policies from the national level to the local administrative levels, through a One Health perspective as follows:

1. Give your point of view about leptospirosis prevention and control of Mahasarakham province;

2. Give your point of view about the cooperation between departments in leptospirosis prevention and control of Mahasarakham province;

3. What are the supporting factors of the success of leptospirosis prevention and control in Mahasarakham province?;

4. What are the factors constituting obstacles to leptospirosis prevention and control in Mahasarakham province?;

5. What are your suggestions?

All conversations were audio-recorded and handwritten notes were taken.

### 3.2. Participants Observation

Meetings including the interviewees from the local levels (3.11) and other representants of the public health and livestock departments from the district level were organized to discuss the implementation of policies for disease prevention and control at the province level. The point was to reflect collectively on the answers given during the interviews on the existing policies. Participant observations were collected during three meetings relating to disease prevention and control in the province of Mahasarakham and one fieldwork in leptospirosis investigation in 2017.

Observations were recorded in the form of field notes to provide a descriptive account of the highlights of the program implementation. Analytical comments were added progressively as insights were gained throughout the dynamic of the discussions and the investigations. The field notes were dated, then entirely transcribed.

### 3.3. Ethical Statement

Approval notices for interviews were given by the Ethics Committee of Mahasarakham University reference number 172083/2021.

### 3.4. Data Analysis

Audio-recordings, notes taken during interviews and participant observations were transcribed verbatim by a dedicated transcriber the day after data collection. The transcript and observation notes were read several times to get a sense of the entire data. The transcripts did not include any personal identifiers to protect the anonymity of the participants. Validation of data analysis through methodological triangulation from different data sources was used to build a coherent justification for the themes. Data analysis involved content analysis in which transcriptions were screened for relevant information which focused on developing coding categories according to the statements and the relevance for the study by identification of key themes and interpretation of the findings within the context.

## 4. Results

### 4.1. The Mechanism of Public Health Implementation in Thailand

The Ministry of Public Health establishes the main health policy and supports academic health knowledge to the public health officers in the province. With the movement of decentralization in Thailand, the Government decided that each provincial governor would be Chief Executive Officer (CEO) administering all activities within his/her jurisdiction on an integrated manner. The health activities in districts vary according to the district health plan. They analyze the main health policy and health problems in the district to devise a district health plan. The public health officers in the Tambon Health Promotion Hospital have an important role in encouraging health activities in the Tambon, i.e., sub-district, which are assessed and controlled by the District Health Officer, the Provincial Public Health Office and the Regional Health Office. In addition, the Department of Public Health cooperates with the Ministry of Interior in order to evaluate indicators and set a budget. The relations between Public Health Departments in Thailand are available in Figure 1.

### 4.2. Evolution of Leptospirosis Prevention and Control Policy Implementation in Mahasarakham

#### 4.2.1. Leptospirosis Prevention and Control Policy Implementation before 2000

In 1996, the first leptospirosis patient was reported at a time when public health staff did not have knowledge on leptospirosis. Thus, the Ministry of Public Health organized training to educate about cause, symptoms and published and provided guidelines for disease prevention to the Regional Public Health Office. The Regional Health Office was then charged to train the Provincial Public Health Office. The knowledge was also provided to small groups [17]. Public health officers were the only persons in charge of leptospirosis prevention and control.

#### 4.2.2. Leptospirosis Prevention and Control Policy Implementation 2000 to 2003

In 2000, the province of Mahasarakham was chosen by the Ministry of Public Health as a pilot province for a better control of the outbreak of leptospirosis and thus for implementing a leptospirosis prevention policy. The province received budget support from the Miyasawa project for conducting prevention and control leptospirosis activities. Mahasarakham Provincial Public Health Office implemented a big campaign for the prevention and control of leptospirosis targeting the population. The government allocated a budget for buying rat tails from people with the aim to encourage people to eradicate rats [17]. A tail is a proof that a rat has been killed: people receive money according to the number of tails presented.

From 2000 to 2003, the Ministry of Public Health supported various media such as brochures or leptospirosis handbooks in order to educate people, community leaders, village health volunteers about leptospirosis prevention and control in the province [18]. During this phase, the Ministry of Public Health and the Ministry of Education cooperated for the promotion of leptospirosis prevention and control [19]. Moreover, a “war room” was established to handle leptospirosis [20].

#### 4.2.3. Leptospirosis Prevention and Control Policy Implementation from 2004 to 2010

After 2003, leptospirosis prevention and control activities became a regular process for public health officers. The prevention and control procedures activities are activated when a found leptospirosis case is declared. The Bureau of Epidemiology established policy for prevention and disease control by creating the Surveillance and Rapid Response Team (SRRT). SRRT consists of approximately five persons including field epidemiology and public health staffs [21]. Mahasarakham province adopted prevention and control diseases policy by developing and implementing SRRT in every district. Prevention and control leptospirosis followed the implementation from Ministry of public health called 4E2C (Early detection; Early diagnosis; Early treatment; Early control; Coordination; Community involvement) as shown in Table 2.

In 2006, the Thai government decided a decentralization reform in the health sector and allocated 24.1% of the central budget dedicated to public health to the Local Government Organisations (LGOs) of the whole country. These LGOs play an important role to develop social services in several forms (roads, water supply, waste management, …), in line with local administration laws.

#### 4.2.4. Leptospirosis Prevention and Control Policy Implementation from 2011 to 2019

In 2011, the Department of Disease Control of the Ministry of Public Health established the prevention disease standard “The District Strengthening Disease Control”. The Department of Disease Control designated districts as target areas for developing disease surveillance and preparing response when there is an outbreak in the district.

Leptospirosis prevention and control of Mahasarakham Province was implemented following the District Strengthening Disease Control which was the standard in preventing disease in the district. The District Strengthening Disease Control has been used from 2011 to now (2019). Leptospirosis prevention and control consist of two strategies via establishing the leptospirosis network participation and Surveillance and Rapid Response. Leptospirosis network participation is activated for leptospirosis prevention while Surveillance and Rapid Response team intervene when a leptospirosis case has been reported. The establishment of a leptospirosis network participation consists of five procedures shown in Figure 2.

Prevention and control of leptospirosis following 4E2C procedure has been established by the Ministry of public health [15]. Currently, prevention and control of leptospirosis are carried out by following the new national Communicable Diseases Law, 2015. The law does not detail the measures to be taken but one of the missions of the government agencies in the province is to jointly plan control and prevention of the disease. The Communicable Diseases Committee of the province meets every 4 months for the planning and the follow-up of their activities. After the meeting, the relevant agencies proceed to their operation plan for prevention and control disease.

#### 4.2.5. Leptospirosis Prevention and Control in Mahasarakham Province, One Health Perspective

The One Health concept was applied for the first time in 2012 in Thailand to address practical health problems. The One Health concept has been adopted as a key component of the national health policy, in particular the National Strategic Plan for Emerging Infectious Diseases (2013–2016) adopted in 2012 [22] and the National Strategic Plan for Emerging Infectious Disease Preparedness, Prevention and Response (2017–2021).

In 2013, four ministries (the Ministry of Public Health, the Ministry of Agriculture and Cooperatives, the Ministry of Natural Resources and Environment and the Ministry of Education) signed the Memorandum of Understanding (MoU) on the implementation of the One Health Initiative for National Health Security. In addition, in 2016, the cooperation between Ministries has been developed further with the signature of an MoU on the joint implementation of the One Health Initiative for National Health Security jointly by seven ministries and one organization [23].

In 2015, the Ministry of Public Health encouraged province prevention and control disease by using the One Health concept. They encouraged the provinces by providing funding through research projects. Nowadays, the use of the One Health approach for the prevention and control of diseases depends on the willingness of the practitioners and on the specific health problems in each province. If a province detects emerging diseases, it must ensure prevention and control, following the National Strategic Plan for Emerging Infectious Disease Preparedness, Prevention and Response. However, other infectious diseases prevention and control measures decided at the district level should follow “The District Strengthening Disease Control”.

Currently, although there is a Communicable Diseases committee of the province, the committee is not specifically interested on the leptospirosis issue at the provincial level. There is no plan or action to foster cooperation between departments of the province for the prevention and control of leptospirosis. At the provincial level, leptospirosis is a priority only for the Public Health Department but not for other departments such as Livestock Department or Agriculture Department. There are guidelines for cooperation between departments following “The District Strengthening Disease Control” adopted before Thailand decided to use the One Health approach.

## 5. Discussion

### 5.1. Evolution of Leptospirosis Prevention Policy Implementation

Thailand has progressively developed a leptospirosis prevention and control policy framework at the national level and transferred the responsibility of its implementation to the local level. Leptospirosis prevention and control is under the responsibility of public health officers.

Before 2000, the policy implementation in this phase was not clearly framed at the local level because the officers lacked understanding of the disease and did not have means to disseminate knowledge to the population.

From 2000 to 2003, the Ministry of Public Health has chosen Mahasarakham as a pilot province to develop and improve leptospirosis prevention policy implementation. Thus, Mahasarakham Provincial Public Health Office supported public health officers in their responsibility to control the disease and gave them a clearer role in the prevention of the disease. A specific budget was dedicated to prevention and control of the disease. The department of public health was the main organization to run prevention and control activities. It developed cooperation with the Department of Education for the education of students in the schools.

From 2004 to 2010, leptospirosis prevention and control has been enhanced at the district level. The cooperation between public health officers in the district is now clearly established and public health officers must share knowledge and resources amongst the subdistricts (tambon) within their district. Although, a movement of decentralization in the Health Sector allocated a proportion of the central budget to the LGO, there was no specific obligation to work on health issues and administrators of LGO were not interested in this sector. Moreover, the public health department worked jointly with the Livestock department only in the case of an outbreak to investigate the disease.

From 2011 to 2014, the leptospirosis prevention disease activities were designed by public health officers at the district level. The committee of Communicable Diseases of province created at that time became an important mechanism making all sectors aware of health issues by developing a comprehensive plan involving the different sectors. Moreover, the committee helped reducing redundancy in the operations driven by each unit. Local government was in charge of allocating the budget for disease prevention in sub districts. Therefore, prevention and control disease activities were more comprehensive and could contribute to reduce the problem of budget sharing. At the provincial level, Public Health Department and Livestock Department were cooperating for the prevention and control of epidemic even before the promotion of the One Health concept. However, the cooperation stayed informal at that time: for a specific mission in case of zoonotic disease outbreak for example. Some prevention and control activities were under the sole responsibility of public health officers and thus there was a clear lack of linkages necessary for comprehensive disease prevention activities. However, after Thailand supported and adopted the One Health approach, the various departments had a better understanding of the importance of cooperation for epidemic prevention and control.

### 5.2. Lessons Learnt for Leptospirosis Prevention Policy Implementation

Prevention and control of leptospirosis needs to be supported by complex scientific knowledge and health policy development. There are complex leptospirosis risk factors such as socioeconomics [24,25], environment [26], livestock [27,28] and community involvement. Departments in districts cooperate to develop plans to prevent and control leptospirosis that mainly focus on patients and not the whole population. However, there are insufficient linkages between departments to ensure comprehensive disease prevention activities at the local level. Strengths and weaknesses in leptospirosis prevention and control policy implementation are shown in Table 3.

The One Health approach, thanks to disease surveillance, management, and eradication through collaboration between public health improving personal protection, veterinarians dealing with livestock and wild animal populations, ecologists and public health experts can improve the prevention and control of the disease [28]. The implementation of the One Health approach can enhance communication on risk and help turning rhetoric into reality [29]. National health policies and systems should allow a better cooperation between departments in accordance with the One Health approach. Moreover, at the local level, a standard joint action should be developed to prevent and control the zoonosis with the participation of local departments and people. This would help to clarify the role of each department for a comprehensive disease prevention and control, consistent with the issues encountered in the local context.

## 6. Conclusions

Leptospirosis outbreak in Mahasarakham Province is still a health issue, although the leptospirosis prevention and control seemed to have gained from the enhanced collaboration between the Public Health Department, the Livestock Department and the Tambon Administration Organization following the prevention strategy of 2011 [30]. However, there is a lack of linkages between departments to ensure comprehensive disease prevention activities. A better implementation of the One Health approach for leptospirosis prevention and control is needed to develop clearer concrete cooperation between provincial departments, and to establish realistic leptospirosis guidelines at the local level. Enhancing the One Health leptospirosis surveillance implementation may also prove beneficial for the control and prevention of other zoonotic diseases.

## Figures and Tables

**Figure 1 tropicalmed-06-00168-f001:**
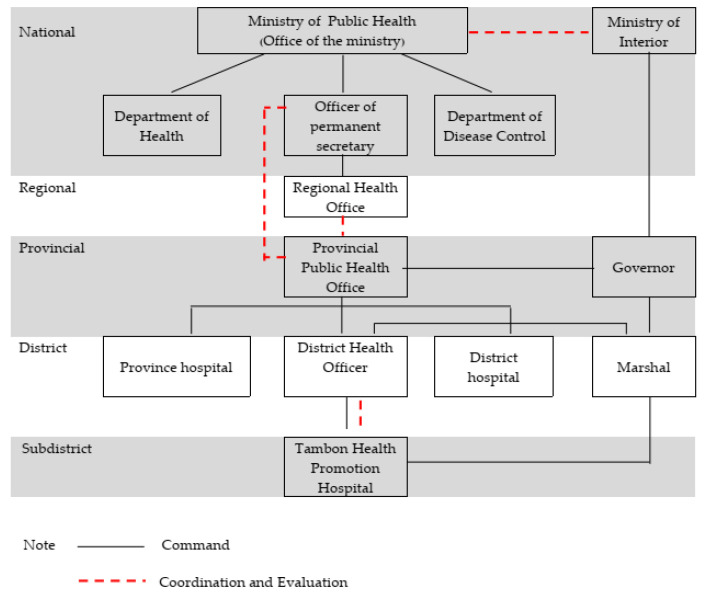
The network of public health departments in Thailand.

**Figure 2 tropicalmed-06-00168-f002:**
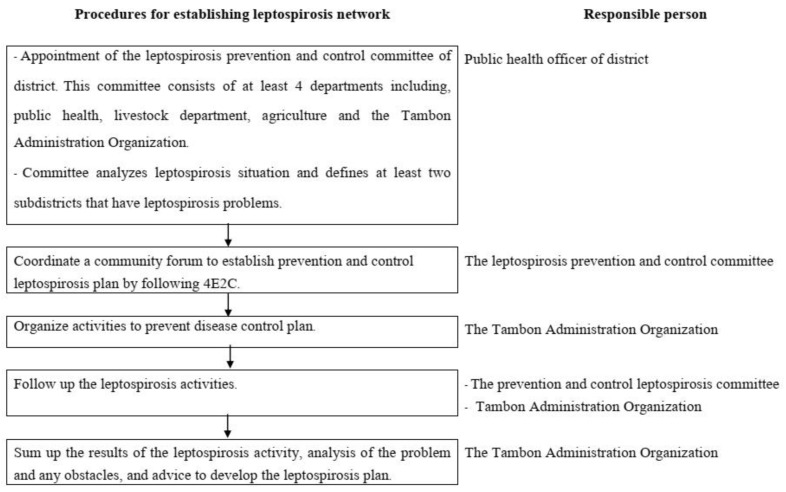
Procedures for establishing leptospirosis network.

**Table 1 tropicalmed-06-00168-t001:** Presentation of study samples.

Group of Population	Number	Data Collection	Sampling
Leader of department of disease control	1	Individual in-depth interviews	-
Officer of coordination unit for One Health	1	Individual in-depth interviews	-
Provincial Public health officers	3	Individual in-depth interviews	purposive sampling
District Public health officers	3	Individual in-depth interviews	purposive sampling
Subdistrict Public health officers	13	Structured interviews	Purposively selected in sub districts with the highest leptospirosis incidence within a district
Local government officers	3	Individual in-depth interviews	purposive sampling
Livestock officers in Mahasarakham	2	Individual in-depth interviews	purposive sampling

**Table 2 tropicalmed-06-00168-t002:** Implementation of the 4E2C tool of the Ministry of Public Health.

Implementation	Activities	Responsible Person
Early detection	Share outbreak information with healthcare providers	Patients Public health volunteers
Early diagnosis	Make a preliminary diagnosis from the patient’s history and symptoms such as acute fever, headache, muscle pain following the Leptospirosis diagnosis guideline	Healthcare providers
Early treatment	Give treatment to patient, following the leptospirosis diagnosis guideline.	Healthcare providers
Early control	Prevention and control of leptospirosis one week after receiving the leptospirosis report.	Surveillance and Rapid Response Team
Coordination	Investigate causes of leptospirosis cases and prepare advice for the people in the community. For example, advise about feeding animal, monitoring animal diseases and reporting leptospirosis cases.	Public health officers Livestock department
Community involvement	Community participates in the prevention and control of leptospirosis. For example, warning about leptospirosis, establishing the leptospirosis prevention in the community etc.	Community

**Table 3 tropicalmed-06-00168-t003:** Strengths and weaknesses in leptospirosis prevention and control policy implementation.

Strengths		
National Level	Provincial Level	Sub-District Level
There is a National Strategic Plan for Emerging Infectious Disease Preparedness, Prevention and Response. There are mechanisms for cooperation between departments via Officer of coordination unit for One Health.	There is a Communicable Diseases Committee in the Province.	There are guidelines for the District Strengthening Disease Control.
**Weaknesses**		
**National Level**	**Provincial Level**	**Sub** **-** **District Level**
No plan and indicator between departments for prevention and control leptospirosis.	The Communicable Diseases Committee is not concerned by leptospirosis issue. The cooperation is only occasional. For example, public health and livestock officer join only to investigate leptospirosis disease punctually.	No standard or clear role for cooperation between departments No indicator to work using the One Health approach Leptospirosis prevention mainly focuses on patients and not the whole population.

## Data Availability

Data sharing not applicable.

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
