# Peer review of "Evolution of Public Health Prevention of Leptospirosis in a One Health Perspective: The Example of Mahasarakham Province (Thailand)"

_tropicalmed, 2021, doi:10.3390/tropicalmed6030168_

Round 1

Reviewer 1 Report

Revision of a manuscript ID tropicalmed-1222255

Title: Evolution of Public Health prevention of leptospirosis in a One Health perspective: the example of Mahasarakham province (Thailand)

            Generally: the revised manuscript is very well written.

            In details:

1) The introduction is concise and gives enough information about the studied problem worldwide. Moreover, the importance of the problem is emphasized. The aim and objective of the study are clearly defined.     

2) About the part “Materials and Methods”:

The Study Design is appropriate for achievement of the aim of the study. The subpart Study population is clear and is well highlighted by the Table 1. Note: On the last row of the Table 1 the cell of total number of people interviewed is empty – my recommendation is either delete the last row of the Table 1 or write 26 – it is total number of people interviewed.

Recommendations: Part 2. Materials and Methods should be written as           2. Materials

Methods should be written as 3.Methods with subheadings:

3.1. Data collection

3.1.1. Steps of data collection

3.1.2. In-depth and structured interviews

3.2. Participants observation

3.3. Ethical statement

3.4. Data analysis

This recommendation is only for improvement of the structuring. The content of this part is clear and precisely described. The methods used are relevant to the design of the study.

3) In the section “Results” in subparts are described The mechanism of Public health implementation in Thailand and Evolution of Leptospirosis prevention and control policy implementation in Mahasarakham. The second subpart is impressive historical survey of the mentioned aspects. In two figures and one table are mentioned all findings about the studied problem. The formatting of the figures is appropriate. The order of the figures and the table follows the logical structure of the study.

4) In the section “Discussion” the authors precisely analyzed the influence of all studied factors upon the incidence of leptospirosis in Mahasarakham Province of Thailand. They discussed the Lessons learnt for leptospirosis prevention policy implementation. On Table 3 they mentioned the Strengths and weaknesses in leptospirosis prevention and control policy implementation on national, provincial and sub-district levels. This facilitates better understanding of the present situation of the studied problem.

5) The conclusion is completely supported by the results of the study.    

6) In the list of References are included 29 sources of information (10 of them from last five years – 34%). They are in order of mentioning in the text and all of them are cited correctly.

            At the end of my notes I want to say that this study is a very good example for “One Health approach” and deserves to be published. It is focused on important aspect of the Public Health and will be of strong interest to the readers of the journal.  

My final recommendation is to accept the manuscript in present form.

Reviewer 2 Report

The work is very difficult to read, and in many parts the text is complicated and incomprehensible to the reader. In general, I have the impression of information chaos. The paper requires redrafting and for sure should be shortened. In my opinion, the work is not suitable for publication at the moment. I suggest publishing this paper at your local journal

Reviewer 3 Report

In lines 76+, the authors state that "The objective of this analysis is to determine the necessary elements of policy, particularly those integrating the One Health approach, which will help to enhance the efficiency of the prevention, detection and response to leptospirosis outbreaks."  I don't think the 'necessary elements' have been adequately explored or discussed.  One Health includes human health, animal health and environmental health - was there any involvement of environmental health experts, and were any of them interviewed?

The section on Data Analysis is extremely brief - this needs to be expanded the provide details of the qualitative analyses and how information from interviews were analysed and summarised to produce the results presented in the paper. 

The conclusion includes a few overall statements about leptospirosis and One Health, but some of these are not conclusions of the current study, or supported by evidence generated by the study.

Round 2

Reviewer 2 Report

The authors made an effort to make corrections to the paper, which does not change my opinion that the work may be interesting but for a region where the problem of leptospirosis is present. I think that the work should be published locally, but I leave the decision to  the editors.

Author Response

Thank you for your comment. This paper is submitted to be included to a Special Issue entitled: Leptospirosis in Humans, Animals and the Environment—A “One Health” Perspective.

Reviewer 3 Report

The Methods section has been improved. It's not clear what section 3.2 Participants observations was about.  How was this different to section 3.1. on interviews?  

Author Response

Thank you for your useful comment.

We added a short explanation about the aim of the meetings in 3.2.

“Meetings including the interviewees from the local levels (3.1.2) and other representants of the public health and livestock departments from the district level were organized to discuss the implementation of policies for disease prevention and control decided at the province level. The point was to reflect collectively on the answers given during the interviews on the existing policies. Participant observations were collected during three meetings relating to disease prevention and control in the province of Mahasarakham and one fieldwork in leptospirosis investigation in 2017”.